# Enterococci, from Harmless Bacteria to a Pathogen

**DOI:** 10.3390/microorganisms8081118

**Published:** 2020-07-25

**Authors:** Sónia Ramos, Vanessa Silva, Maria de Lurdes Enes Dapkevicius, Gilberto Igrejas, Patrícia Poeta

**Affiliations:** 1Microbiology and Antibiotic Resistance Team (MicroART), Department of Veterinary Sciences, University of Trás-os-Montes and Alto Douro (UTAD), 5001-801 Vila Real, Portugal; soniacatarinaramos@gmail.com (S.R.); vanessasilva@utad.pt (V.S.); 2Department of Genetics and Biotechnology, University of Trás-os-Montes and Alto Douro, 5001-801 Vila Real, Portugal; gigrejas@utad.pt; 3Functional Genomics and Proteomics Unit, University of Trás-os-Montes and Alto Douro (UTAD), 5001-801 Vila Real, Portugal; 4Associated Laboratory for Green Chemistry (LAQV-REQUIMTE), University NOVA of Lisboa, 2829-516 Lisboa, Caparica, Portugal; 5Faculty of Agricultural and Environmental Sciences, University of the Azores, 9500-321 Angra do Heroísmo, Portugal; maria.ln.dapkevicius@uac.pt; 6Institute of Agricultural and Environmental Research and Technology (IITAA), University of the Azores, 9500-321 Angra do Heroísmo, Portugal

**Keywords:** *Enterococcus*, antibiotic resistance, vancomycin resistance, public health, nosocomial opportunists

## Abstract

Enterococci are gastrointestinal commensals whose hardiness allowed them to colonize very diverse environments, including soils, water, food, and feed. This ability to overcome adverse conditions makes enterococci problematic once they colonize hospital niches. Together with the malleability of their genomes, the capacity to acquire and disseminate determinants of antibiotic resistance has contributed to converting what was once just another opportunistic pathogen into a first-class clinical problem. This review discusses the dimension of the emergence of enterococcal resistance to key antimicrobial agents, the dissemination of this resistance, and its significance in terms of public health, with the aim of raising awareness of the need to devise and implement surveillance programs and more effective antibiotic stewardship.

## 1. Introduction

Traditionally, enterococci were regarded as a harmless commensal bacterium, and were even believed to have positive effects on a number of gastrointestinal and systemic conditions. However, when the commensal relationship with the host is disrupted, enterococci can cause invasive infections [1]. Enterococci are Gram-positive catalase-negative, non-spore-forming, facultative anaerobic lactic acid bacteria, and normal inhabitants of the gut flora of humans, many different mammals, birds, fish, reptiles, amphibians, and insects, as well as nematodes [2,3]. Until 1984, the enterococci were considered as part of the genus *Streptococcus*, but they have constituted a unique taxonomic entity since the mid-1980s [4,5]. Today, over 50 different species of enterococci have been described, of which *E. faecium* and *E. faecalis* are the most common in the human gastrointestinal tract, whereas, among farm animals *E. faecium* together with *E. cecorum*, *E. faecalis* and, to some extent, *E. hirae* predominate, while *E. mundtii* and *E. casseliflavus* are commonly found in plant sources [6,7]. Moreover, ecology and epidemiological studies have reported *E. faecalis* and *E. faecium* as frequently isolated from food products (cheese, fish, sausages, minced beef, and pork) and the environment (sewage, soil, and water) [8]. Due to their preferred intestinal habitat, their wide occurrence, robustness, and ease of cultivation, enterococci are used as indicators of fecal contamination and are part of the hygiene standards for water and food products. Additionally, they are also suitable as important key indicator bacteria for veterinary and human resistance surveillance systems [5].

Because they produce bacteriocins, enterococcal isolates have a long-standing tradition as starter cultures or as supplements in food fermentation and food preservation [9]. Bacteriocin ST15 from *E. mundtii* has been shown to be effective against a range of Gram-positive and Gram-negative bacteria including *Acinetobacter*, *Bacillus*, *Clostridium*, *Klebsiella*, *Lactobacillus,* and *Pseudomonas*, whereas enterocins A, B, I, L and P, are active against *Listeria* species, *Clostridium* species, and *S. aureus* [10]. Another beneficial aspect is the use of *E. faecalis* as a probiotic, promoting a positive gut environment. Additionally, enterococci have been shown to strengthen the immune system, reduce inflammation, and may even be indirectly involved in reducing the incidence of colon cancer [11].

Enterococci are very hardy organisms; they can sustain various adverse conditions and survive for several months in the environment [6]. They are able to survive in a range of stressful and hostile environments, including extreme conditions of pH and temperatures (between 10 °C and 45 °C), plus high NaCl concentration [8]. These attributes make enterococci ideally suited for fermentation applications but, ironically, these same attributes make them difficult to eliminate and control once they become established in a hospital environment.

Enterococci are now firmly established as one of the major nosocomial pathogens and are increasingly becoming more resistant to antimicrobial agents. Presently, almost all nosocomial enterococcal infections are caused by, either, *E. faecalis*, or *E. faecium* [6]. Of these, *E. faecalis* is the most pathogenic species, but *E. faecium* is of increasing importance as, in general, it frequently is more resistant to antimicrobials [4]. Commonly, these organisms are involved in hospital-acquired infections such as catheter-associated urinary tract infections, endocarditis, bacteremia, neonatal sepsis, surgical and burn wound infections, and more rarely meningitis [6].

Enterococci are typically harmless in healthy individuals. They become opportunistic pathogens mainly by causing infections in patients who are in Intensive Care Units, who suffer from a severe underlying disease, or who are immunocompromised. Therefore, the severity of illness and immune suppression can be directly associated with prolonged hospital and/or indiscriminate antibiotics use, and these are major risk factors for nosocomial acquisition of drug-resistant enterococci [12]. Antibiotic treatment may create new niches and nutrient resources, wherefore the existing commensal gut microbiota can be eliminated and subsequently replaced by opportunistic pathogens [13]. Patients in hospitals are typically treated with broad-spectrum antibiotics (penicillins and cephalosporins), which dramatically increases hospital-associated *E. faecium* colonization in their small intestine, cecum, and colon, outcompeting the normal Gram-negative gut microbiota [14,15]. In addition, some antibiotic therapies may lower the levels of C-type lectin RegIII produced by the host, allowing *E. faecium* to overgrow in the intestinal tract [15]. Furthermore, the intrinsic resistance of enterococci to several commonly used antibiotics and, perhaps more importantly, their malleable genomes plus their capacity to acquire and disseminate determinants of antibiotic resistance, are major factors that may have contributed to their adaptation to harsh environments [16]. Both microbial and host factors can contribute to the conversion of a second-rate pathogen into a first-rate clinical problem, as is seen nowadays.

## 2. Emergence and Dissemination Trends of Vancomycin-Resistant Enterococci (VRE) Strains

Vancomycin-resistant enterococci (VRE) have spread with unanticipated rapidity and today is steadily increasing worldwide. However, European countries and the United States (US) have experienced differences in VRE emergence and epidemiology [17]. In the 1990s, the rapid emergence of VRE observed in the US was preceded by the emergence of ampicillin-resistant *E. faecium* in the early 1980s, while in Europe, the first VRE clinical isolates were only detected in 1986 [18,19]. In the U.S., colonization of hospitalized patients with VRE rapidly increased since the first VRE detection, up to the current endemic levels in many hospitals. On the contrary, in Europe, the prevalence rates in hospitals have remained much lower and only started to increase since the year 2000 [17,20]. The initial reports of VRE in Europe were of organisms that were frequently isolated from healthy people, farm animals, pets, and retail food products, suggesting a large community reservoir [14,21], whereas such a community reservoir seemed absent in the US. Until recently, the detection of VRE from food-production animals in the US was infrequent and the first report of VRE in food animals was published only by 2010 [22].

These epidemiological differences between the US and Europe presumably resulted from the abundant antibiotic use in US hospitals, most notably of vancomycin and cephalosporins, where the emergence of VRE was preceded by the emergence of ampicillin-resistant enterococci, making them more susceptible to the selective effects of antibiotics [6,19]. In Europe, it has been suggested that the extensive and widespread use of avoparcin (a vancomycin-like glycopeptide never used in the US) in animal husbandry was associated with the high numbers of VRE in animal feces and meat samples, which subsequently, could have colonized healthy humans via the food chain, thus explaining the initial community reservoir [8]. Furthermore, the detection of VRE in hospitalized individuals, when they had not previously been admitted to a hospital or taken antibiotics, suggests that VRE may have been contracted through the food chain [10].

When the potential hazard of avoparcin use to the VRE emergence was recognized, its use was banned. After the withdrawal of avoparcin, the prevalence of VRE in farm animals in Europe rapidly declined [8]. On the other hand, and despite the fact glycopeptide resistance had already declined in community reservoirs, subsequent surveillance showed that the European continent has continued to experience an important increase in the isolation of VRE, especially *E. faecium*, in hospitals [14]. Indeed, more recently, the European Antimicrobial Resistance Surveillance System (EARSS) reported that the prevalence of clinical vancomycin resistance in *E. faecium* increased throughout Europe from 10.5% in 2015 to 17.3% in 2018 while vancomycin-resistant *E. faecalis* remained low in most countries [23]. Nevertheless, large variations between different European countries are seen, with VRE ranging from <1% (France, Iceland, Luxembourg, and Slovenia) to >50% (Cyprus) [23].

Concerning animal prevalence, and despite avoparcin being withdrawn for more than 15 years, vancomycin resistance is still detected in enterococci isolated from food-producing animals, albeit at low to very low levels. Nevertheless, VRE are found in the intestinal microbiota of farm animals in Europe, whereas in the USA carriage of VRE is usually absent [24]. Different theories about why VRE persist among farm animals have been presented. Studies from Denmark and Norway showed that the use of other antimicrobial growth promoters may lead to a coselection phenomenon, and reduced VRE numbers were only documentable when other growth promoters (spiramycin, tylosin) were also banned, as both resistance determinants *erm*(B) and *van*A may be located on similar plasmids [4,25]. Another explanation that has been suggested is that plasmid addiction systems, “selfish” DNA sequences located on the same plasmid as the *van*A gene, would force the bacteria to retain the resistance by killing the plasmid-free cells [4]. Resistance to tetracyclines and erythromycin is also commonly detected among the indicator enterococci isolates from animals and food [24]. Further, in a few European studies, putative linkage of glycopeptide, macrolide, and tetracycline-resistant genes have been implicated in the occurrence of VRE in the feces of food-producing animals [26,27].

## 3. The Public Health Impact of VRE

Historically, *E. faecalis* has caused the majority of all enterococcal infections (80–90%), however, lately, the proportion of *E. faecium* infections has increased surpassing the prevalence of *E. faecalis* [28,29]. This may be due to the penetration of ampicillin-resistant *E. faecium* isolates in hospitals, whereas resistance to ampicillin and vancomycin in *E. faecalis* remains less common [16]. With the partial replacement of *E. faecalis* by *E. faecium* as a cause of enterococcal infection and the simultaneous rapid epidemic rise of multiresistant *E*. *faecium* clones, dramatic epidemiological changes have been felt in hospitals all over the world during the last two decades [16].

Because enterococci are “tough bugs” that can survive for long periods on environmental surfaces and are tolerant to heat, chlorine, and some alcohol preparations, their control is highly challenging once established in a hospital environment [16]. Nowadays, VRE infections are increasingly common and difficult to treat, appearing usually as long-lasting hospital outbreaks that represent tremendous difficulties for infection control [30]. Moreover, patients are considered a major reservoir and fecal carriers of VRE strains. Hence, in hospital settings, VRE dissemination, and transmission can occur through direct contact with colonized or infected patients, or through indirect contact via the hands of healthcare workers, or via contaminated equipment such as thermometers (especially rectal thermometers), bedrails, gloves, and environmental surfaces [28]. The hospital environment plays, indeed, a key role in the nosocomial dissemination of VRE. The attachment of VRE to and their long-term persistence in inanimate surfaces [31,32], as well as their resistance to standard cleaning procedures [33], may create a reservoir from which complex nosocomial dissemination routes may spread these pathogens. Investigating these dissemination routes is of utmost importance to design and evaluate contention measures [34]. Active screening has been reported to reduce the rates of VRE infection [5]. However, approaches based on the finding of environmental sampling do not always yield enough information to establish the contamination routes and study the dynamics of VRE outbreaks [35]. Whole-genome sequencing has been increasingly used as a tool to provide insights into the nosocomial spread of VRE [35,36,37,38]. The introduction of routine WG as an additional screening tool in hospitals would, therefore, provide valuable information for the design of effective infection control strategies [38].

Infection prevention and control strategies may be classified in environments focused (such as cleaning and disinfection procedures, use of antimicrobial materials for inanimate surfaces, use air filters and negative pressure rooms), and human-focused (antimicrobial stewardship, hand hygiene, patient decolonization, and educational actions) [39,40]. Hand hygiene, chlorhexidine bathing, environmental cleaning protocols, and antimicrobial stewardship have been identified as the main components for prevention and control of VRE in hospital settings [34], whereas prevention of human colonization, patient decolonization and the use of a “One Health” approach has been proposed as the most promising novel strategies for an integrated hospital and community control effort of these opportunists [39,40].

The treatment of VRE severe infections is difficult, non-conventional, and demands the use of antibiotic combinations. VRE infections are associated with a higher mortality rate and economic burden compared to those caused by vancomycin susceptible enterococci [41]. In addition, the rise in prevalence of enterococcal infections in humans may be influenced, to some degree, by their acquired and intrinsic resistance mechanisms towards the most commonly used antibiotics. All *Enterococcus* species show intrinsic resistance to antimicrobial agents such as ß-lactams, aminoglycosides (low level), and vancomycin (low level, in *E. gallinarum* and *E. casseliflavus*) [8]. Enterococci do not possess penicillin-binding proteins (PBPs), which bind cephalosporins with high affinity and, as a result of the poor permeability of the enterococcal cell wall, aminoglycosides are unable to reach their target site [8]. For a long time, the treatment of choice in serious enterococcal infections was the synergistic effect of penicillin/ampicillin or vancomycin and an aminoglycoside. The combination of ampicillin (or vancomycin) and gentamicin create sufficient injury to the cell wall to provide uptake and exposure to aminoglycosides, creating a bactericidal synergism [19,42]. However, the combination of high-level resistance to ampicillin, vancomycin, and aminoglycosides is now common with hospital-acquired *E*. *faecium*. Furthermore, such strains also commonly carry resistance genes to other classes of antimicrobials, such as fluoroquinolones, macrolides, and tetracyclines [7]. Hence, therapeutic alternatives towards these multiresistant VRE infections are restricted to antibiotics recently introduced into clinical practice such as quinupristin/dalfopristin, linezolid, tigecycline, and daptomycin. However, these drugs are only approved for certain situations and resistance has already been reported [8,43]. Furthermore, the widespread emergence of linezolid resistance in *E. faecalis* and *E. faecium* [44,45,46,47,48], which has been attributed to various mechanisms, such as mutations in the V domain of the 23S rRNA [44,45,46,47,48], in the genes encoding the L3, L4 and L22 riboproteins [48] or by genetic determinants carried in plasmids (*cfr*, *optrA,* and *poxtA*) [46,47,48], may pose additional challenges to the treatment of infections by these agents.

Enterococci capacity to acquire and disseminate determinants of antibiotic resistance contributed largely to the actual epidemiological situation. The most common transmissible elements in enterococci are the Tn*3* family transposons, like Tn*917* (conferring resistance to MLSB antibiotics) and Tn*1546* (conferring glycopeptide resistance) and the conjugative transposon Tn*916*, which confers resistance to minocycline and tetracycline [49]. Furthermore, vancomycin resistance among enterococci most probably spread via the dissemination of variants of the vanA-type element Tn*1546*, mostly located on mobilizable or conjugative plasmids [50]. Among *E. faecalis* sex-pheromone inducible plasmids, carrying resistance to tetracycline, aminoglycoside and glycopeptide have also been identified [5], whereas the broad host range plasmids of the Inc18 group were found in *E. faecium* isolates. These are associated with resistance to MLSB, aminoglycosides, chloramphenicol, tetracycline, and glycopeptides. Moreover, Inc18-type plasmids have been implicated in the transfer of vancomycin resistance determinants to *S. aureus* [49,51].

Given the ease of acquisition of resistance genes by the enterococci, the prudent use of antibiotics as therapeutic agents is an important tool in the management of the antibiotic resistance problem [5]. Antibiotic stewardship programs, especially those that involve prospective auditing and feedback, have been shown to have an important impact on the resistance rates in individual institutions [52,53]. They reduced the overall usage of these drugs, thereby decreasing the selective pressure on the hospital microbiota, and increased the heterogeneity of the prescribed antimicrobial agents [52,54]. Both the European Center for Disease Control (ECDC) and the Center for Disease Control (CDC) have guidelines regarding antimicrobial stewardship programs [55,56]. CDC has identified hospital leadership commitment, accountability, pharmacy expertise, action, tracking, reporting, and education as the core elements an antibiotic stewardship program should comprise [56]. 

Along with enterococci malleable genome, their ability to the acquisition of virulence factors may have also contributed to their nosocomial prevalence [57]. The genome sequence of *E. faecalis* strain V583, revealed the striking fact that over 25% of its genome is made up of a mobile and exogenously acquired DNA, including a large pathogenicity island encoding known virulence traits [58]. In enterococci, the presence of virulence factors can enhance their pathogenicity by allowing the colonization and invasion of host tissue, translocation through epithelial cells, and evasion from the host’s immune response [57].

Putative virulence factors such as aggregation substance, adhesins, hemolysin, hyaluronidase, and gelatinase, play an important role in establishing infection. The aggregation substance (Agg) is a pheromone-inducible surface glycoprotein that mediates aggregate formation during conjugation thus, assisting in plasmid transfer, as well as, adhesion to an array of eukaryotic surfaces [59]. Whereas Ace, a collagen-binding protein, belonging to the Microbial Surface Components Recognizing Adhesive Matrix Molecules (MSCRAMM) family, is considered to be important in the early stages of infection, as they may bind components of the host extracellular matrix [60]. The surface protein (Esp), in addition to a role in adhesion, may also have a function in evasion from the host’s immune response and in antibiotic resistance [60]. Esp also contributes to biofilm formation, which could have a role in experimental UTI and/or endocarditis models [60]. The hydrolytic enzyme hyaluronidase, encoded by the chromosomal *hyl* gene, acts on hyaluronic acid and is associated with tissue damage. The secreted virulence factor, cytolysin (also called hemolysin), a bacterial toxin produced by ~30 % of *E. faecalis* strains, enables the bacteria to evade the host immune response by destroying cells such as macrophages and neutrophils [61]. Moreover, it has ß-hemolytic properties in humans and is bactericidal against other Gram-positive bacteria [61]. Other important secreted factors include the proteases gelatinase (GelE) and the serine proteinase (SprE). The main role of these proteases in enterococcal pathogenesis is thought to be in providing nutrients to the bacteria by degrading host tissue. In addition, GelE seems to have an important role in clearing misfolded proteins. The genes encoding these proteases are regulated by the Fsr quorum sensing system [61].

Furthermore, MLST-based studies have shown that strains isolated from hospital-acquired infections clustered together in specific groups, termed clonal complexes (CC). Particularly, CC2 and CC9 in *E. faecalis* and, CC17 in *E. faecium*, have been associated with nosocomial infection [62]. The relative increase in the proportion of *E. faecium* to *E. faecalis* has been associated with the spread of a particular hospital-adapted polyclonal high-risk enterococcal complex (HiRECC) *E. faecium* CC17 [63]. This specific *E. faecium* subpopulation is mostly characterized by ampicillin-resistance, high-level ciprofloxacin-resistance, and to possess an accessory genome, which, includes putative virulence traits such as the *esp* gene, the hyaluronidase *hyl_Efm_*, and the *acm* gene (encoding a collagen-binding protein) [63].

There is limited evidence as to the direct role of the food-producing animals in the dissemination of VRE among humans. However, this potential hazard has been widely recognized. Identical Tn*1546* variants among VRE isolates have been recovered from food-producing animals and humans, indicating a common human and animal reservoir for *van*A elements [17,27]. Moreover, strains from human-adapted CCs causing most enterococcal infections may eventually be recovered from farm and pets (e.g., *E. faecium* CC17 and *E. faecalis* CC2), and strains from CCs commonly found among animals have also been isolated from humans (e.g., *E. faecium* CC5, *E. faecalis* ST16 or CC21) [21]. Also, Freitas et al. [11] were able to demonstrate clonal relationships between hospital- and swine-associated VRE, indicating that this route of transmission does exist [61].

## 4. Persistence of VRE Strains in Portuguese Food Animals, an Example

Vancomycin is still a frequently used antibiotic to treat infections caused by multiresistant enterococci, despite the present expansion of VRE strains representing a tremendous challenge to human infection control. In Europe, as previously explained, it has been suggested that the massive use of avoparcin in animal husbandry was associated with the high prevalence rates of VRE strains detected in food-producing animals and their subsequent expansion into the community. Once the use of avoparcin was discontinued, the prevalence of VRE among farm animals decreased. 

Nevertheless, VRE are still present among farm animals. Figure 1 compares the prevalence of VRE reported in different settings in Portugal (food-producing animals, environmental, wastewater treatment plants, wild animals, and pets). The displayed data show that VRE strains are still broadly distributed in Portugal, being isolated not only from healthy food-producing animals, wild animals, and pets, but also from the environment. VRE are found in wastewater treatment plants and in pig breeding facilities. The environmental prevalence of VRE is troublesome, as the release of effluents into watercourses and the use of sludge in agriculture might actively contribute to the dissemination of VRE strains, resistant bacteria and resistance genes throughout the environment [64].

Furthermore, the persistence of VRE in food-producing animals and related environments (years after avoparcin withdrawal) indicates that coselection with other antimicrobial agents increased fitness of strains and the presence of specific mobile genetic elements cannot be ruled out. It is known that the *erm*B gene, encoding for macrolide resistance, can be carried by the same conjugative plasmid harboring *van*A gene [73]. Moreover, a putative linkage of the glycopeptide, macrolide, and tetracycline-resistant genes has been implicated in the occurrence of VRE in the feces of food-producing animals [16]. In fact, tetracycline and macrolides are widely prescribed in pig husbandry to control respiratory and enteric disease and the use of one of these antimicrobials may favor the spread of resistance against antimicrobial from different groups. In a recent study, all VRE strains with acquired mechanisms of resistance (VREar) from pigs showed coresistance to tetracycline and erythromycin, which supports the hypothesis that this linkage could be a causative factor for the ongoing persistence of VRE [26]. It is important to notice that the mobile element Tn*916*/Tn*1545*-like transposon was detected in the majority of our VREar strains. This is consistent with the results from other Portuguese settings, where these mobile genetic elements were also frequently associated with acquired vancomycin resistance [67,71]. Moreover, it was reported, in Portugal, that human and swine share vancomycin-resistant *E. faecium* strains harboring Tn*1546* on indistinguishable plasmids [21]. In addition, the rapid and extensive spread of VRE in Portuguese hospitals seems to be associated with the dissemination of the *van*A gene on Tn*1546*-type transposons [21]. Nonetheless, VRE persistence, being continuously reported among animals and the environment, shows that this data should not be overlooked and must be continuously monitored.

In addition to food-producing animals, VRE and their resistance genes have been reported and detected in foods at retail (meat, vegetables, cheese, and milk) [74,75,76,77]. This is a cause for enormous concern since it favors the dissemination of antimicrobial resistance organisms, consequently reducing therapeutic options [78]. In Portugal, only a few studies have reported the presence of VRE in food such as cheese, poultry carcasses, and processed meat [74,79,80,81,82,83,84].

At present, our knowledge regarding the occurrence of antimicrobial resistance in food-producing animals, the quantitative impact by the use of different antimicrobial agents on the selection of resistance, and the most appropriate treatment protocols to limit the development of resistance still have some limitations. Prevalence of antimicrobial resistance in studies on food-producing animals contributes towards establishing a knowledge base regarding the emergence of resistant bacteria. Accordingly, programs monitoring the occurrence and development of resistance and consumption of antimicrobial agents are strongly desirable, as is research into the most appropriate ways to use antimicrobial. In turn, this could aid in the implementation of guidelines and regulations on the usage of antimicrobial agents in livestock production systems. Such guidelines for the prudent use of antimicrobial agents may help to slow down the selection for resistance, should be based on knowledge regarding the normal susceptibility patterns of the causative agents, and take into account the potential problems for human health.

## Figures and Tables

**Figure 1 microorganisms-08-01118-f001:**
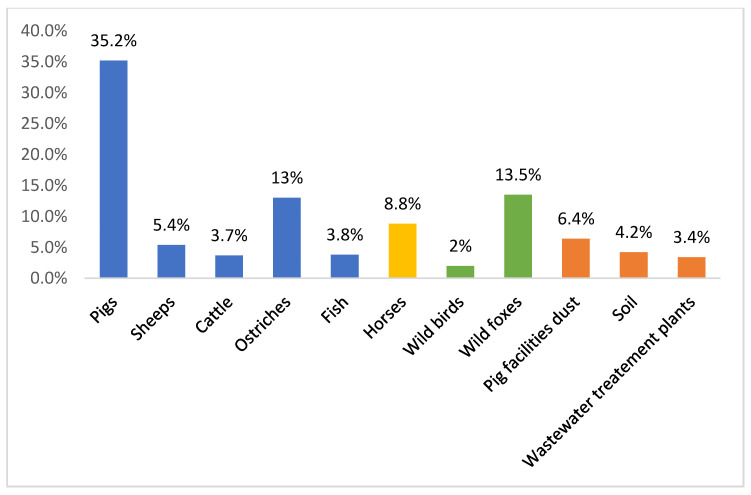
Percentages of vancomycin-resistant enterococci reported in Portugal in different settings: food-producing animals, environmental, wastewater treatment plants, wild animals, horses, and pets [26,65,66,67,68,69,70,71,72].

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
