# Peer review of "Enterococci, from Harmless Bacteria to a Pathogen"

_microorganisms, 2020, doi:10.3390/microorganisms8081118_

Round 1

Reviewer 1 Report

This is a well written, review article, employing appropriate methodology that will contribute to the literature.

Author Response

The authors thank the reviewer for his/her analysis of the manuscript and for the very positive assessment given.

Reviewer 2 Report

Ramos et al. give an overview over the current significance of enterococci inside and outside the hospital setting. The review article is well written, easy to understand and reports the current level of knowledge. However, three important aspects should be included to strengthen the clinical relevance of VRE:

1) There are several (outbreak) investigations concerning VRE in hospitals. Some of these should be discussed, especially regarding environmental VRE-contamination and transmission from inanimate surfaces. Additionally, the role of whole genome sequencing approaches can be discussed in this context.  

2) Please state at least in a few words, which infection control measures in hospitals are recommended and which different perspectives are present at the moment. 

3) In lines 153 ff. the authors give an overview of different therapeutic strategies. Here, the role of antibiotic stewardship experts could also be discussed, as there is evidence for the implementation of such teams in preventing VRE in hospitals.

Author Response

Comments and Suggestions for Authors

Ramos et al. give an overview over the current significance of enterococci inside and outside the hospital setting. The review article is well written, easy to understand and reports the current level of knowledge. However, three important aspects should be included to strengthen the clinical relevance of VRE:

1) There are several (outbreak) investigations concerning VRE in hospitals. Some of these should be discussed, especially regarding environmental VRE-contamination and transmission from inanimate surfaces. Additionally, the role of whole genome sequencing approaches can be discussed in this context.

Although a brief discussion of the role of environmental contamination and transmission had already been provided in lines 151 – 154 of the first version of the manuscript, additional text has been inserted to provide a more comprehensive discussion, as requested. Additional text on the role of WGS has also been added (see lines 151 – 162 of the revised manuscript).

2) Please state at least in a few words, which infection control measures in hospitals are recommended and which different perspectives are present at the moment.

Additional text has been provided regarding infection control measures in hospitals (see lines 163 – 171 of the revised manuscript).

3) In lines 153 ff. the authors give an overview of different therapeutic strategies. Here, the role of antibiotic stewardship experts could also be discussed, as there is evidence for the implementation of such teams in preventing VRE in hospitals.

Additional text has been provided, as requested (see lines 208 – 218 of the revised manuscript).

Submission Date

29 June 2020

Date of this review

01 Jul 2020 11:39:42

Reviewer 3 Report

GENERAL COMMENTS: This short review succintly overviews the emergence of enterococci as commen commensal inhabitatnts of the gut flora to significant pathogens in hospitalized patients in the last 20 years. The review covers many factors that have contributed to the emergence of enterococci as significant hospital pathogens in recent years, including the use of antibiotics in animal feed, the emergence of vancomycin-resistant enterococci and the widespread use of antibiotics in hospitals. In general the review provides a valuable overview of  the emergence of enterococci as hospital pathogens.

I have provided some specific comments below for the authors’ consideration. Thee are intended to be constructive and to improve clarity and impact.

SPECIFIC COMMENTS:

(1). The manuscript is reasonably well written but would benefit enormously from a thorough editing for English use. I very much appreciate that English may not be the authors first language. I have provided a small few exampes below.

Line 26: Change to: ..., with the aim of raising awareness of the need to devise and implement surveillance programmes and more effective antiobiotic stewardship.

Line 81: “same”?  Is “some” intended?

Line 106: “the massive use of avoparcin”. Suggest “the extensive and widespread use of avoparcin”

Line 121: “Island”. Is “Ireland” intended?

(2). Line 130:Please clarify exactly what you mean by “plasmid addiction systems”. This is very unclear.

(3). Line 151: Failure to adequately decontaminate rectal thermometers in the hospital setting is an important consideration for tranmission of VREs. Please mention this specifically.

(4). One of the shortcomings of this review that can easily be rectified is that little attentiion has been given to the enormous impact that whole-genome sequencing (WGS) has had on our understanding of the populatin biology of enterococci. WGS has revolutionized our understanding of enterococcal infection in hospitals and has dramatically enhanced surveillance and surveillance tracking of hospital outbreaks (see doi.org/10.1016/j.jhin.2020.05.013). I  would actively encourage the authors to include a section in the review covering these areas. I appreciate that the review may be constrained by word-count-but nonetheless this would really strengthen the review.

(5). Finally, the recent emergence of linezolid resistance in enterococci has posed additional challenges to treating enterococcal infections. Resistance can emerge by the acquisition of mutations in ribosomal genes and by acquisition of specific genes frequently encoded on conjugative plasmids (e.g. optrA, cfr variants and poxtA). Many of thee genes were originally identified in enterococci in farm animals-but are becoming increasing prevalent in human isolates (see doi:10.1093/jac/dkaa227 &

doi:10.1093/jac/dkaa075).

In my view, this area should be covered by the review as it has a significant bearing on treatment options for hospitalized patients.

Author Response

Comments and Suggestions for Authors

GENERAL COMMENTS: This short review succintly overviews the emergence of enterococci as commen commensal inhabitatnts of the gut flora to significant pathogens in hospitalized patients in the last 20 years. The review covers many factors that have contributed to the emergence of enterococci as significant hospital pathogens in recent years, including the use of antibiotics in animal feed, the emergence of vancomycin-resistant enterococci and the widespread use of antibiotics in hospitals. In general the review provides a valuable overview of  the emergence of enterococci as hospital pathogens.

I have provided some specific comments below for the authors’ consideration. Thee are intended to be constructive and to improve clarity and impact.

SPECIFIC COMMENTS:

(1). The manuscript is reasonably well written but would benefit enormously from a thorough editing for English use. I very much appreciate that English may not be the authors first language. I have provided a small few exampes below.

A thorough language revision has been performed.

Line 26: Change to: ..., with the aim of raising awareness of the need to devise and implement surveillance programmes and more effective antiobiotic stewardship.

The requested change has been made.

Line 81: “same”?  Is “some” intended?

The requested change has been made.

Line 106: “the massive use of avoparcin”. Suggest “the extensive and widespread use of avoparcin”

The requested change has been made.

Line 121: “Island”. Is “Ireland” intended?

 The text has been corrected. It should have read “Iceland” instead of “Island”.

(2). Line 130:Please clarify exactly what you mean by “plasmid addiction systems”. This is very unclear.

Plasmid addiction systems are “selfish” DNA portions that some plasmids contain and that ensure plasmid persistence through successive bacterial generation by post-segregational killing of cells that do not contain the plasmid. A short explanation has been included in the text, as requested (see lines 129 – 121 of the revised manuscript).           

 (3). Line 151: Failure to adequately decontaminate rectal thermometers in the hospital setting is an important consideration for tranmission of VREs. Please mention this specifically.

Mention to rectal thermometers has been included, as requested (see line 151).

(4). One of the shortcomings of this review that can easily be rectified is that little attentiion has been given to the enormous impact that whole-genome sequencing (WGS) has had on our understanding of the populatin biology of enterococci. WGS has revolutionized our understanding of enterococcal infection in hospitals and has dramatically enhanced surveillance and surveillance tracking of hospital outbreaks (see doi.org/10.1016/j.jhin.2020.05.013). I would actively encourage the authors to include a section in the review covering these areas. I appreciate that the review may be constrained by word-count-but nonetheless this would really strengthen the review.

A succinct discussion on WGS has been introduced, as requested (see lines 156 – 162 of the revised manuscript).

(5). Finally, the recent emergence of linezolid resistance in enterococci has posed additional challenges to treating enterococcal infections. Resistance can emerge by the acquisition of mutations in ribosomal genes and by acquisition of specific genes frequently encoded on conjugative plasmids (e.g. optrA, cfr variants and poxtA). Many of thee genes were originally identified in enterococci in farm animals-but are becoming increasing prevalent in human isolates (see doi:10.1093/jac/dkaa227 & doi:10.1093/jac/dkaa075).

In my view, this area should be covered by the review as it has a significant bearing on treatment options for hospitalized patients.

A brief discussion on the emergence of linezolid resistance in enterococci has been added, as requested (see lines 191 – 195 of the revised manuscript).

Submission Date

29 June 2020

Date of this review

09 Jul 2020 09:48:29

Round 2

Reviewer 2 Report

The manuscript substantially improved after revision. From the clinical perspective all relevant points are now included.